

# Functional characterisation of phagocytes in the Pacific oyster *Crassostrea gigas*

Shuai Jiang[1], Zhihao Jia[1], Tao Zhang[1], Lingling Wang[1,2], Limei Qiu[1], Jinsheng Sun[3] and Linsheng Song[2]

[1] Key Laboratory of Experimental Marine Biology, Institute of Oceanology, Chinese Academy of Sciences, Qingdao, Shandong, China
[2] Key Laboratory of Mariculture & Stock Enhancement in North China's Sea, Ministry of Agriculture, Dalian Ocean University, Dalian, Liaoning, China
[3] Tianjin Key Laboratory of Animal and Plant Resistance, Tianjin Normal University, Tianjin, China

## ABSTRACT

Invertebrates lack canonical adaptive immunity and mainly rely on innate immune system to fight against pathogens. The phagocytes, which could engulf and kill microbial pathogens, are likely to be of great importance and have to undertake significant roles in invertebrate immune defense. In the present study, flow cytometry combined with histological and lectin staining was employed to characterise functional features of phagocytes in the Pacific oyster *Crassostrea gigas*. Based on the cell size and cellular contents, haemocytes were categorised into three cell types, i.e., granulocytes, semigranulocytes and agranulocytes. Agranulocytes with smaller cell volume and lower cytoplasmic-to-nuclear ratio did not show phagocytic activity, while semigranulocytes and agranulocytes exhibited larger cell volume, higher cytoplasmic-to-nuclear ratio and phagocytic activity. In addition, granulocytes with higher side scatter (SSC) exhibited higher phagocytic activity than that of semigranulocytes. When $\beta$-integrin and lectin-like receptors were blocked by RGD tripeptide and carbohydrates, respectively, the phagocytic activity of both granulocytes and semigranulocytes was significantly inhibited, indicating that $\beta$-integrin and certain lectin-like receptors were involved in phagocytosis towards microbes. Moreover, lipopolysaccharide but not peptidylglycan could enhance phagocytic activity of granulocytes and semigranulocytes towards *Vibrio splendidus* and *Staphylococcus aureus*. Lectin staining analysis revealed that *Lycopersicon esculentum* lectin (LEL), binding the epitope polylactosamine, was highly distributed on the extracellular cell surface of phagocytes, and could be utilized as a potential molecular marker to differentiate phagocytes from non-phagocytic haemocytes. The results, collectively, provide knowledge on the functional characters of oyster phagocytes, which would contribute to deep investigation of cell typing and cellular immunity in bivalves.

## INTRODUCTION

Phagocytosis, the uptake and digestion of exogenous particles, is an ancient, evolutionarily conserved cellular process, which plays important roles in the pathogen killing and clearance as well as the food uptake (*Aderem & Underhill, 1999*; *Greenberg & Grinstein, 2002*; *Henneke & Golenbock, 2004*). In mammals, professional phagocytes, such as macrophages and

Corresponding author
Linsheng Song, lshsong@qdio.ac.cn, lshsong@dlou.edu.cn

neutrophils, are particularly responsible for the killing and the clearance of pathogens, and for initiating signaling pathways to that trigger potent immune responses (*Kaufmann & Dorhoi, 2016*). More importantly, some phagocytes named antigen-presenting cells (APCs) are well known not only for their potent phagocytic activity but also for the antigen presentation activity, which has been deemed as the bridge between innate immunity and adaptive immunity (*Delamarre et al., 2005*).

Invertebrates, which lack the canonical adaptive immunity based on B and T lymphocytes, mainly rely on the innate immune system to fight against pathogens (*Little, Hultmark & Read, 2005*; *Milutinovic & Kurtz, 2016*). Compared with the phagocytes in vertebrates, invertebrate phagocytes are likely to be of great importance and have even major roles in immune defense (*Robb et al., 2014*). In previous studies, haemocytes have been typed into various cell subpopulations in several invertebrates, and some subpopulations have been confirmed to be in charge of phagocytosis. For instance, granulocytes in *Crassostrea virginica* were found to be active in phagocytosis (*Goedken & De Guise, 2004*). In *Drosophila melanogaster*, plasmatocytes were specifically responsible for the ingestion of microorganisms, while lamellocytes and crystal cells were involved in encapsulation and melanisation respectively (*Lemaitre & Hoffmann, 2007*). Recently, the capture and engulfment of bacteria by circulating or fixed phagocytes have also been reported in several invertebrates (*Le Grand et al., 2011*; *Soderhall, 2010*).

The Pacific oyster *Crassostrea gigas* is an important species for physiological ecology as well as economical resource (*Zhang et al., 2012*). In the present study, flow cytometry combined with histological staining was employed to categorise the haemocytes based on their morphological features; the phagocytic activities of different cell populations were also determined. The phagocytic modulation effects of $\beta$-integrin, lectin-like receptors (LLRs), lipopolysaccharide (LPS) and peptidylglycan (PGN) were investigated, and the potential glycan markers distinguishing phagocytes from non-phagocytic cells were screened in order to better understand phagocytosis in oyster innate immune defense.

## MATERIALS AND METHODS

### Animal rearing and manipulation

Oysters, 10–15 cm in length and 150–200 g in weight, were collected from a farm in Qingdao, Shandong Province, China, and acclimated in aerated seawater at 18 °C for two weeks prior to use. All the experiments were conducted according to the regulations of local and central government. The animal experiments were approved by the local animal care and use committee.

### Preparation of haemocytes from *C. gigas*

Haemolymph was withdrawn using a syringe equipped with a needle ($0.9 \times 25$ mm) from the pericardial cavity of adult *C. gigas* specimens after the shells were carefully opened, and mixed immediately with prechilled antiaggregant ACD-A (0.1 mol/l trisodium citrate, 0.11 mol/l dextrose and 71 mmol/l citric acid monohydrate) at a volume ratio of 7: 1. Haemocytes ($\sim 1 \times 10^6$ haemocytes per oyster) were pelleted at 800 g, at 4 °C for 10 min, and washed twice with modified Leibovitz L15 medium (supplemented with 0.54 g/l

KCl, 0.6 g/l $CaCl_2$, 1 g/l $MgSO_4$, 3.9 g/l $MgCl_2$, 20.2 g/l NaCl, 100 units/ml penicillin G, 40 µg/l gentamycin, 100 µg/ml streptomycin, 0.1 µg/ml amphotericin B and 10% fetal bovine serum) (*Novas, Barcia & Ramos-Martinez, 2007*). The Haemocytes from three to five individuals were pooled together as one sample, and $1 \times 10^6$ haemocytes were prepared and stored on ice to reduce spontaneous aggregation. The cell viability was measured using the Trypan Blue exclusion assay (*McCarthy et al., 2014*).

## Phagocytosis assay and May-Grunwald Giemsa (MGG) staining

For phagocytosis assay, haemocytes were incubated with *Pichia pastoris* at a ratio of 1:100 for 1 h, and washed by modified L15 medium for three times. Haemocytes were plated onto glass slides to allow cell adhesion at 18 °C for 3 h, and the glass slides were fixed with 100% methanol for 10 min. MGG was used to stain cells for another 10 min followed by PBS washing, and the cells on the slides were characterized by light microscopy according to their morphological features.

## Preparation of FITC-labeled microbes

*Vibrio splendidus* was grown in 2216E media at 28 °C, at 220 rpm for 12 h. *Escherichia coli*, *Staphylococcus aureus* and *Bacillus subtilis* were grown in LB media at 37 °C, at 220 rpm for 8 h. *Pichia pastoris* was grown in YPD media at 30 °C, at 220 rpm for 24 h. All the microbes were grown to mid-log phase and harvested by centrifugation at 6,000 g for 15 min. Cells were fixed with 4% paraformaldehyde (PFA) for 10 min, washed with 0.1 M $NaHCO_3$ (pH 9.0) for three times, and then mixed with 1 mg/ml FITC (Sigma-Aldrich) in 0.1 M $NaHCO_3$ (pH 9.0) buffer at room temperature with continuous gentle stirring overnight. The FITC-labeled microbes were washed with PBS for three times to eliminate free FITC molecules.

## Flow cytometric analysis of haemocyte and its phagocytosis

Haemocytes were collected and analyzed on a FACS Arial II flow cytometer (Becton Dickinson Biosciences). For morphological characterisation of haemocytes, forward scatter (FSC) combined with side scatter (SSC) analysis was performed to measure relative cell size and internal complexity of cells respectively. For phagocytosis analysis, FITC-labeled microbes and latex beads (Sigma-Aldrich) were incubated with haemocytes at a ratio of 100:1 at 18 °C for 1 h. Cells were then washed by modified L15 medium three times, and Trypan Blue (1.2 mg/ml) was used to quench surface-bound FITC-labeled bacteria. FSC and FL1 channel detection was immediately performed to analyze the phagocytosis of FITC-labeled particles.

## RGD, carbohydrates and PAMPs treatments of haemocytes

Haemocytes were incubated with Arg-Gly-Asp (RGD) tripeptide at 0.5 mg/ml for 1 h to block $\beta$-integrin, and incubated with different carbohydrates including glucose (Glu), fucose (Fuc), mannose (Man), lactose (Lac) and N-acetylglucosamine (GlcNAc) at 100 mM for 1 h to block lectin like receptors (LLRs), respectively. For the LPS and PGN stimulations, haemocytes were incubated with LPS and PGN at 0.1 and 1 mg/ml for 1 h, respectively. Cells were then washed with modified L15 medium for three times, followed by incubation

with FITC-labeled microbes at a ratio of 1:100. Flow cytometry was performed to analyse the percentages of phagocytosing haemocytes.

### Flow cytometric and confocal microscopic analysis of lectin staining

For flow cytometric analysis, haemocytes were incubated with FITC labeled microbes at a ratio of 1:100 at 18 °C for 1 h followed by extensively washing, and then incubated with phycoerythrin (PE)-labeled wheat germ agglutinin (WGA), peanut agglutinin (PNA) and *Lycopersicon esculentum* lectin (LEL) (50 μg/ml) at room temperature for 1 h. After washing with L-15 medium for three times, the haemocytes were analyzed by flow cytometry (BD FACSAria II). For microscopic analysis, haemocytes were collected and suspended in the cell culture medium at the concentration of $1 \times 10^6$ cells/ml. The cell suspension (1.5 ml) was then added in cell culture dishes and incubated for 3 h to allow cell adhesion. Haemocytes were incubated for 1 h with FITC-labeled latex beads at a ratio of 1:100. Haemocytes were then fixed by 4% PFA at 4 °C for 15 min after washing with three times L-15 medium, and permeabilised by 0.1% Triton X-100 for 15 min. Nonspecific binding sites were blocked by adding 5% BSA and incubated at room temperature for 1 h. PE-labeled LEL (50 μg/ml) was incubated with haemocytes at room temperature for another 1 h and washed three times with PBS. Haemocytes were monitored and fluorescent images were taken using Carl Zeiss LSM 710 confocal microscope (Jena, Germany).

### Statistical analysis

The one-way ANOVA followed by Dunnett's multiple comparison test and Student's *t* test were used for the comparisons between groups. Statistical analysis was performed with GraphPad Prism 5 software. The statistical significance was defined as $p < 0.05$.

## RESULTS

### Morphological characters of the haemocytes from *C. gigas*

Haemocytes collected from *C. gigas* were gated by light-scatter characteristics using flow cytometer, and May-Grunwald-Giemsa (MGG) staining was performed to characterize the cellular morphology of each subpopulation (Fig. 1A). Based on the forward scatter (FSC) and side scatter (SSC) intensity, haemocytes were divided into three subpopulations including agranulocytes, granulocytes and semigranulocytes. Agranulocytes were located at the lower left position on the light scatter chart with smaller size (approximate 5–8 μm), clear cytoplasm and lower cytoplasmic-to-nuclear ratio. Granulocytes were located at the upper right position with larger cell size (approximate 10–14 μm), abundant intracellular contents, and higher cytoplasmic-to-nuclear ratio. Semigranulocytes were located at the lower right position with larger cell size (approximate 11–13 μm ), lower internal complexity, and higher cytoplasmic-to-nuclear ratio (Fig. 1A). In addition, most agranulocytes and granulocytes appeared approximately round-shaped on the glass slide, while some of semigranulocytes extended filopodia to explore the microenvironment and spread on the glass slide. A total of 10, 000 haemocytes were analyzed by flow cytometry, and the agranulocyte, granulocyte and semigranulocyte subpopulations comprised 46.2%, 31.4% and 19.6% of the total haemocytes, respectively (Fig. 1B).
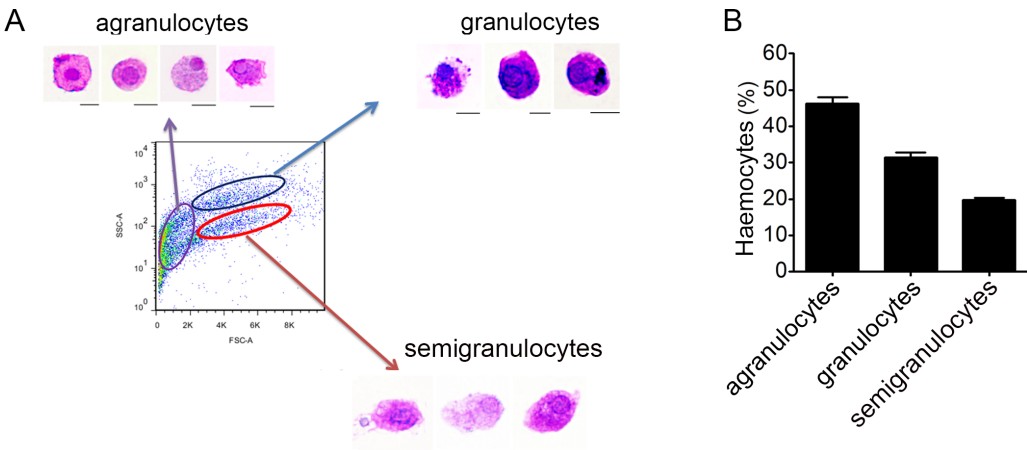

**Figure 1  Flow cytometric analysis and May-Grunwald-Giemsa (MGG) staining of haemocytes from** ***C. gigas.*** (A) Haemocytes were categorized into different subpopulations by flow cytometry followed by MGG staining analysis. Bar: 5 μm. (B) The percentages of each subpopulation were calculated with statistically analysis. Results are means ±S.E.M. ($n = 6$).

## Morphological identification of phagocytes from *C. gigas*

In order to gain a further observation of phagocytes, *Pichia pastoris* with large cell diameter was employed as exogenous particles to allow phagocytosis of haemocytes, and MGG staining was performed to characterize the histological features of phagocytes (Fig. 2A). Phagocytes exhibited larger cell diameter after phagocytosing (approximately 9–14 μm) with higher cytoplasmic-to-nuclear ratio (engulfment of 4–7 fungal cells per phagocyte), while non-phagocytic cells exhibited smaller cell size (5–9 μm), and their nucleus almost filled the cell, leaving a thin rim of cytoplasm. The morphological features of phagocytes were further characterized by flow cytometric analysis (Fig. 2B), and these cells with engulfment of FITC-labeled latex beads were featured with larger cell size (higher FSC value). The percentage of phagocytes in total haemocytes was calculated to be 8.86%. The haemocytes with smaller cell size (lower FSC value) did not exhibit phagocytic capability towards FITC-labeled latex beads. Although the phagocytes ingesting yeast cells could be different from the haemocytes ingesting latex beads, the results collectively suggested that phagocytic haemocytes are probably represented by granulocytes and semigranulocytes, which possess larger cell size and higher cytoplasmic-to-nuclear ratio.

## Involvement of *β*-integrin in phagocytosis

Phagocytes are in charge of phagocytizing exogenous particles, and the phagocytic capability of oyster haemocytes towards different microbes was further investigated by flow cytometric analysis. The percentages of the phagocytic haemocytes were 24.8% for *E. coli* (Fig. 3A), 8.2% for *V. splendidus* (Fig. 3B) and 14.7% for *S. aureus* (Fig. 3C). Moreover, granulocytes exhibited higher phagocytic percentages than that of semigranulocytes, which were 38.7% and 19.1% for *E. coli*, 9.8% and 7.4% for *V. splendidus,* and 24.1% and 10.3% for *S. aureus*, respectively. The total percentages of the phagocytic haemocytes significantly decreased of 43.2% for *E. coli*  (Fig. 3A), 39.6% for *V. splendidus* (Fig. 3B) and 45.7% for

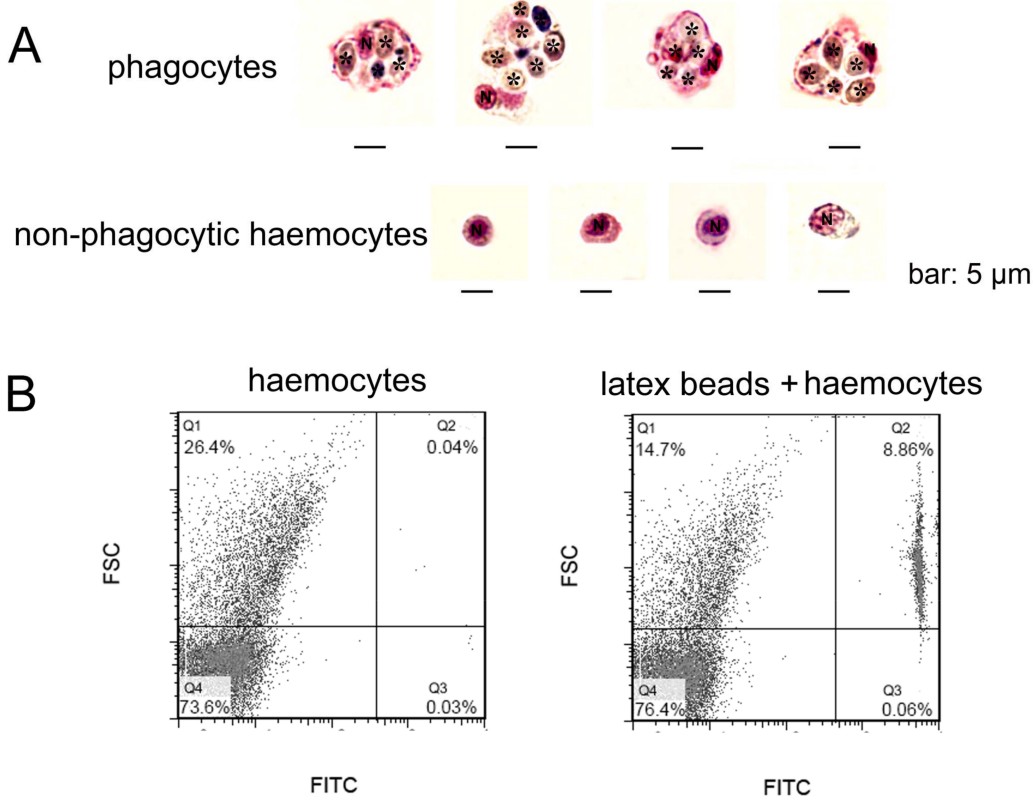

**Figure 2** **Morphological identification of phagocytes from *C. gigas*.** (A) Haemocytes were incubated with fungal cells *Pichia pastoris* to allow phagocytosis, followed by MGG staining and microscopic analysis. Fungal cells are indicated with asterisks, N stands for cell nucleus. Bar: 5 μm. (B) Haemocytes preincubated with FITC-labeled latex beads (2 μm diameter) and analyzed by flow cytometry.

*S. aureus* (Fig. 3C) after the incubation with RGD. Moreover, under the same conditions, the phagocytic percentages of granulocytes and semigranulocytes significantly decreased of 37.8% and 46.4% for *E. coli*, 35.1% and 47.8% for *V. splendidus*, 40.5% and 48.4% for *S. aureus*, respectively.

## The involvement of lectin-like receptors in the phagocytosis of different microbes

The participation of LLRs in phagocytosis towards microbes was determined after the incubation with Glucose (Glu), fucose (Fuc), mannose (Man), lactose (Lac) and N-acetylglucosamine (GlcNAc), respectively. Fuc, Man and GlcNAc exhibited significantly inhibitory effects on the phagocytosis of haemocytes towards *V. splendidus* with the percentages of phagocytosing haemocytes decreased to 58.4%, 67.1% and 64.6%, respectively, while Glu and Lac did not show any significant inhibition on phagocytosis of total haemocytes towards *V. splendidus* (Fig. 4A). After the treatments with Fuc, Lac and GlcNAc, the percentages of phagocytosing haemocytes decreased to 73.4%, 75.5% and 72.2% in granulocytes (Fig. 4B), and the percentages of phagocytosing semigranulocytes decreased to 54.8% and 52.4% after Fuc and Man treatments, (Fig. 4C). In addition, Fuc, Man and GlcNAc exhibited inhibitory effects on the phagocytosis towards *S. aureus*

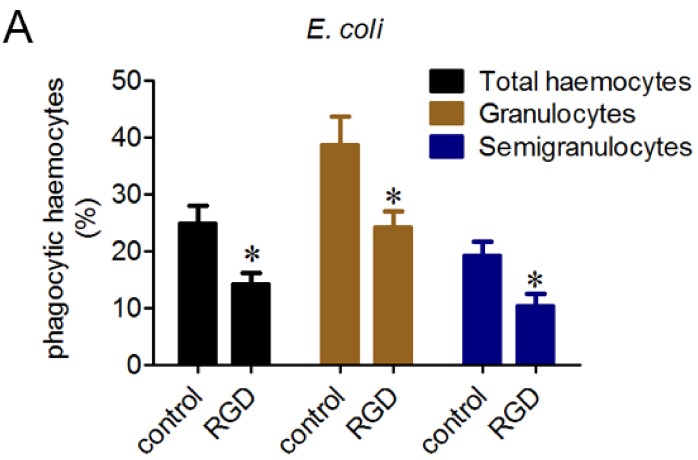

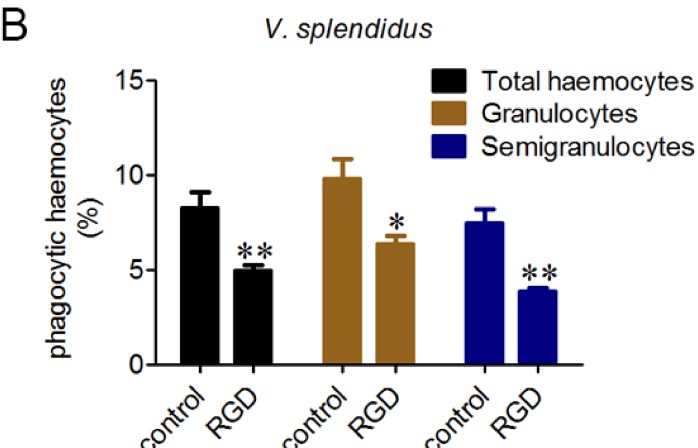

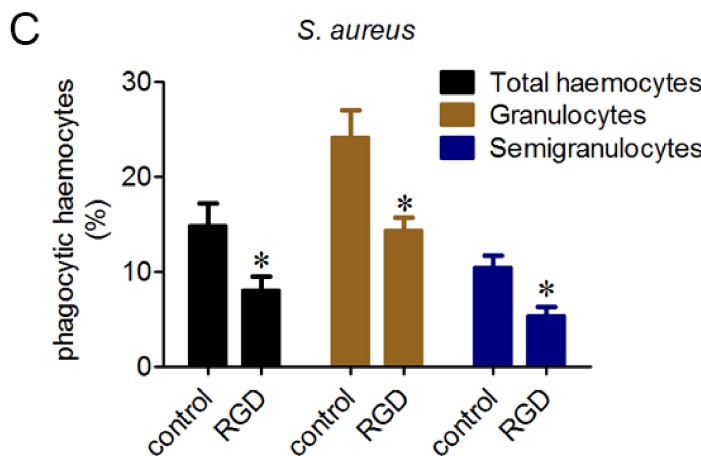

**Figure 3** **Involvement of $\beta$-integrin in phagocytosis towards different microbes.** Haemocytes were treated with or without RGD tripeptide, and the phagocytic activities towards *E. coli*, *V. splendidus* and *S. aureus* were determined by flow cytometry. Phagocytic percentages were shown respectively (A, B and C). Results are means $\pm$ S.D. ($n = 6$), $^*p < 0.05$, $^{**}p < 0.01$.

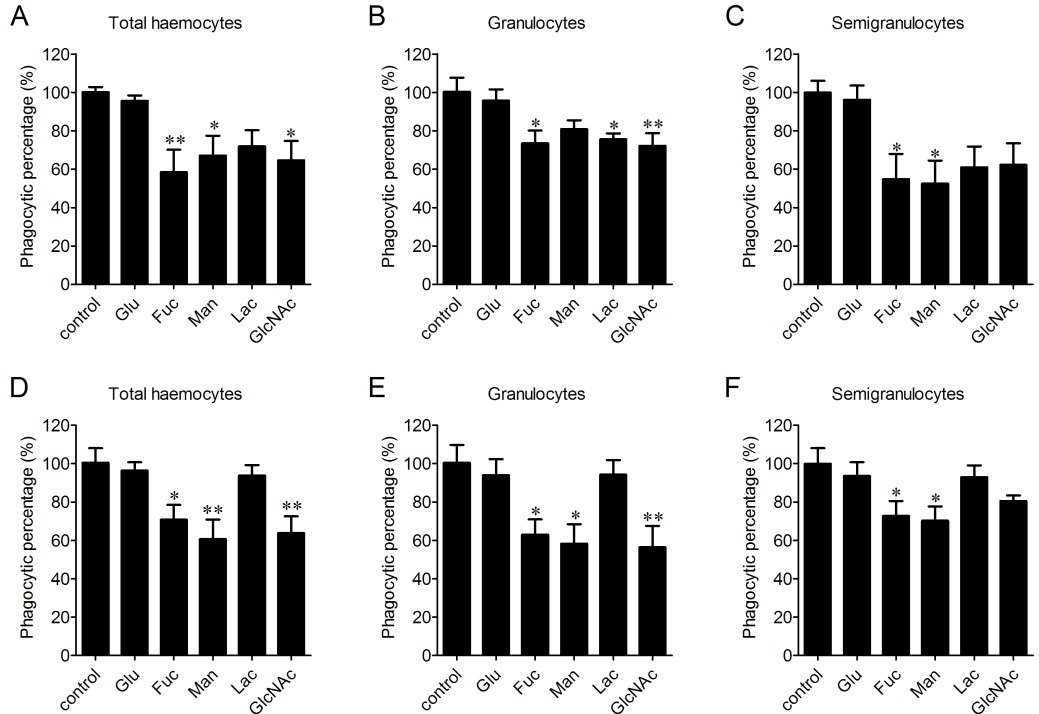

**Figure 4  Involvement of lectin-like receptors (LLRs) in the phagocytosis of different microbes.** Haemocytes were pre-incubated with different carbohydrates, and the phagocytic inhibitory activity towards *V. splendidus* (A, B, C) and *S. aureus* (D, E, F) was determined. Results are means ± S.D. ($n = 6$), *$p < 0.05$, **$p < 0.01$.

in total haemocytes, and the percentages of phagocytosing haemocytes significantly decreased to 70.7%, 60.5% and 63.7%, respectively, while the inhibitory effects of Glu and Lac on phagocytosis of total haemocytes towards *S. aureus* were much lower (Fig. 4D). The percentages of phagocytosing granulocytes towards *S. aureus* were also significantly decreased to 62.9%, 58.1% and 56.4% after Fuc, Man and GlcNAc treatment, respectively (Fig. 4E), and the percentages of phagocytosing semigranulocytes towards *S. aureus* decreased to 72.7% and 70.2% after Fuc and Man treatments (Fig. 4F).

## The enhancement of phagocytosis after LPS treatment

LPS and PGN are important pathogen-associated molecular patterns (PAMPs) identified from Gram-negative and Gram-positive bacteria, respectively. The percentages of phagocytosing haemocytes towards *V. splendidus* increased of 17.8% and 44.3% after 0.01 and 0.1 mg/ml LPS stimulation (Fig. 5A). Meanwhile, it increased of 11.5% and 18.9% in granulocytes (Fig. 5B), and of 25.4% and 53.6% in semigranulocytes, respectively (Fig. 5C). Similarly, LPS stimulation significantly increased the percentages of phagocytosing haemocytes towards *S. aureus* of 16.8% and 31.6% in total haemocytes after 0.01 and 0.1 mg/ml LPS stimulation (Fig. 5D), of 14.2% and 29.3% increment in granulocytes (Fig. 5E), and of 19.5% and 34.2% increment in semigranulocytes, respectively (Fig. 5F).

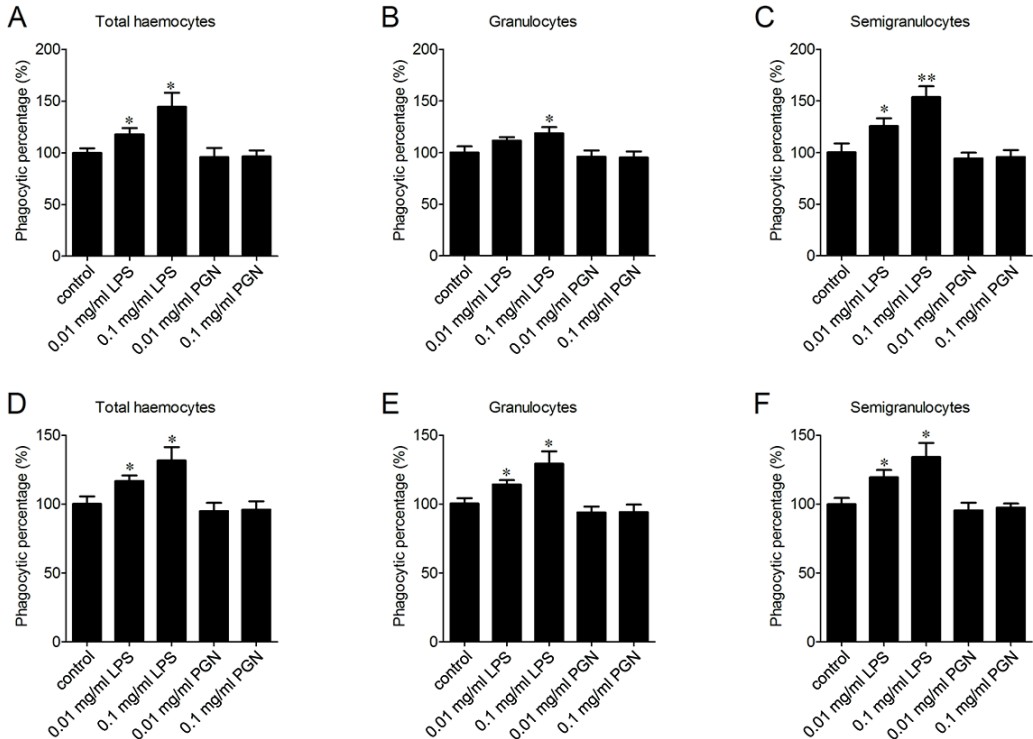

**Figure 5** **Enhancement of phagocytosis after LPS treatment.** Haemocytes were pre-treated with LPS and PGN at different concentrations, and the phagocytic activities towards *V. splendidus* (A, B, C) and *S. aureus* (D, E, F) were determined. Results are means ± S.D. ($n = 6$), $^*p < 0.05$, $^{**}p < 0.01$.

By contrast, there were no significant changes in the percentages of phagocytosing haemocytes towards *V. splendidus* and *S. aureus* in the total haemocytes, granulocytes or semigranulocytes after 0.01 and 0.1 mg/ml PGN treatments, respectively.

### *Lycopersicon esculentum* lectin exhibited high binding specificity to phagocytes

In order to further characterize the molecular features of phagocytes, lectin staining was performed to distinguish phagocytes from non-phagocytic cells (Fig. 6A). Cells positive to *Lycopersicon esculentum* lectin (LEL) staining were in high accordance with the phagocytes from *C. gigas*. The percentages of double positive cells (PE-LEL$^+$/FITC$^+$) were approximately 23.2% for *E. coli*, 18.7% for *V. splendidus*, 24.1% for *B. subtilis* and 27.6% for *S. aureus*, while the percentages of PE-LEL$^-$/FITC$^+$ cells were no more than 3% for all the four microbes, and the percentages of PE-LEL$^+$/FITC$^-$ cells were approximately 3.2% for *E. coli*, 4.1% for *V. splendidus*, 2.7% for *B. subtilis* and 3.4% for *S. aureus*, respectively (Fig. 6B, right).

Conversely, the haemocytes positive to wheat germ agglutinin (WGA) and peanut agglutinin (PNA) staining were not significantly associated with phagocytes. The percentages of PE-WGA$^+$/FITC$^-$ cells were even higher than that of PE-WGA$^+$/FITC$^+$ cells, indicating that WGA exhibited binding activity to both phagocytes and non-phagocytic haemocytes (Fig. 6B, left). Additionally, PE-PNA$^-$/FITC$^+$ cells exhibited

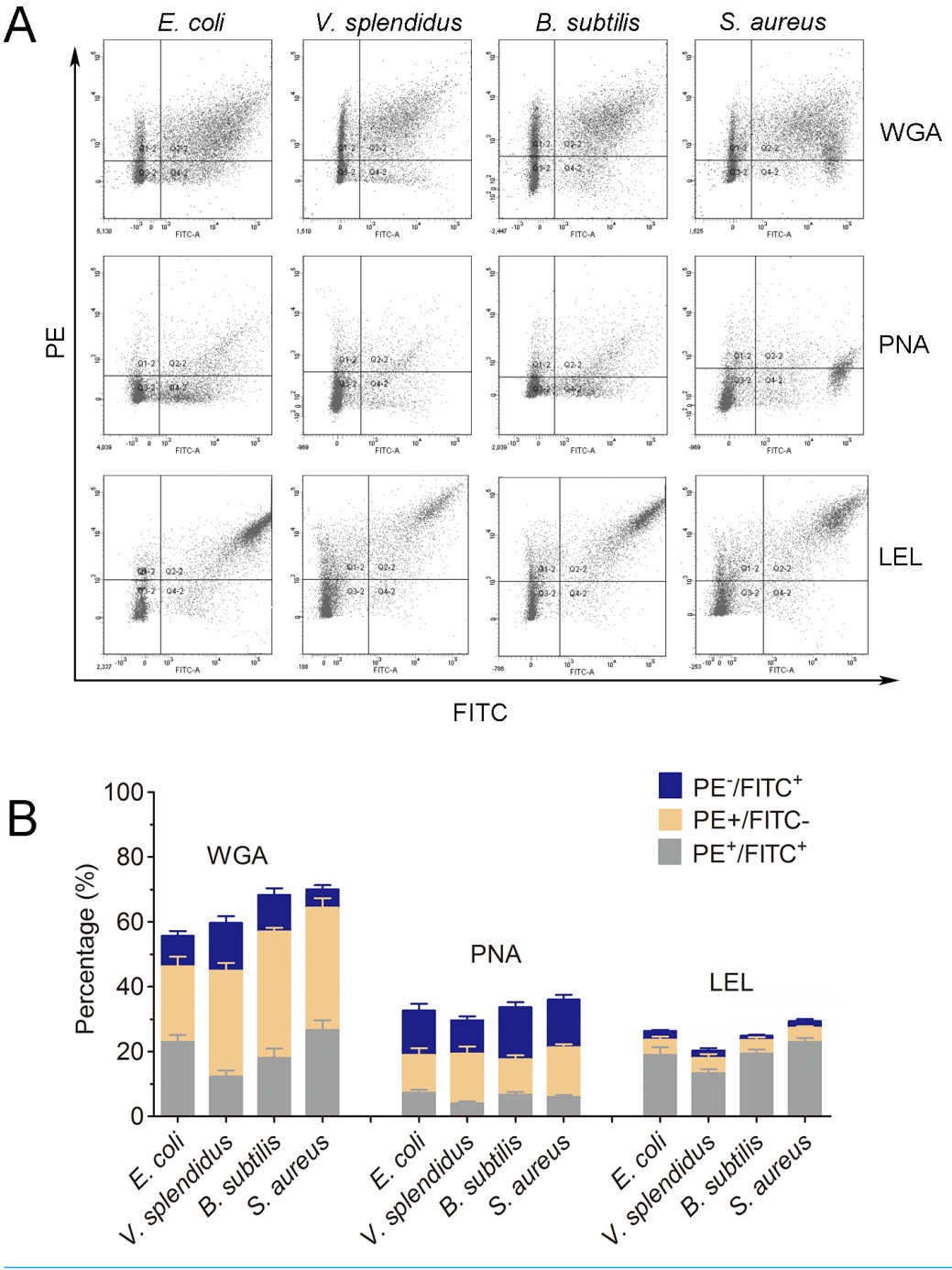

**Figure 6** **Lectin staining analysis of the phagocytes from *C. gigas*.** (A) Haemocytes incubated with FITC-labeled *E. coli*, *V. splendidus*, *S. aureus* and *B. subtilis*, and stained with PE-labeled WGA, PNA and LEL respectively. The correlation between lectin staining and phagocytes was analyzed by flow cytometry. (B) Percentages of haemocytes gated on PE$^+$/FITC$^+$, PE$^+$/FITC$^-$ and PE$^-$/FITC$^+$ ($n = 5$).

higher percentages than that of PE-PNA$^+$/FITC$^+$ cells, suggesting that PNA preferred binding to non-phagocytic haemocytes rather than phagocytes (Fig. 6B, middle). The results clearly indicated that LEL positive staining cells exhibited higher accordance with phagocytes, while WGA and PNA staining exhibited much lower binding specificity towards phagocytes and non-phagocytic cells.

## The distribution of polylactosamine in oyster phagocytes

The LEL binding carbohydrate epitopes, including N-linked and O-linked polylactosamines, are reported with at least three lactosamine repeats in Fig. 7A. The phagocyte-specific distribution of polylactosamine was further confirmed by confocal microscopic analysis. Polylactosamine, as indicated by PE-labeled LEL (red color), was highly distributed on the cell membrane, and assembled to form a cap on one side of the phagocytes in the merge confocal image (Fig. 7B, merge/phagocyte panel). It was noted that polylactosamine also concentrated as patches in cytoplasm of phagocytes. By contrast, there was no positive signal of LEL in non-phagocytic haemocytes (Fig. 7B, merge/non-phagocyte cell panel), indicating that polylactosamine probably not expressed in non-phagocytic cells.

## DISCUSSION

Phagocytes are in charge of phagocytosis, encapsulation and oxidative killing, and provide the main effectors to kill pathogens and sustain immune homeostasis (*Pham et al., 2007*). In previous studies, various criteria were applied to haemocyte classification in bivalve molluscs. For example, it was proposed to divide the haemocytes from *C. gigas* into several groups including basophilic and eosinophilic granulocytes, blast-like haemocytes and large basophilic agranular haemocytes (*Bachere, Chagot & Grizel, 1988*; *Hine, 1999*). Different morphs of haemocytes of *Crassostrea rhizophorae* were proposed to correspond to different developmental stages of the same cell type, which accumulated or lost granules and complexity in response to environmental or microbial challenges (*Rebelo Mde et al., 2013*). Moreover, a granular population composed of basophilic and eosinophilic granulocytes in oysters was reported to possess phagocytic activity (*Bachere et al., 2004*). In the present study, we divided the oyster haemocytes into three cell subpopulations by flow cytometry based on cell size and intracellular contents: agranulocytes, granulocytes and semigranulocytes. Both granulocytes and semigranulocytes exhibited phagocytic activity towards FITC-labeled latex beads and different microbes, while the agranulocytes with smaller cell size did not exhibit phagocytic activity. In addition, the granulocytes exhibited higher phagocytic activity than that of semigranulocytes. It has been reported that the granules in oyster haemocytes were rich in antimicrobial peptides, which were reported to bound specifically to phagosomes and be rapidly released into the phagosomes/phagolysosomes to kill the phagocytosed microbial pathogens, suggesting a vital role of granulocyte in the clearance of pathogen (*Gonzalez et al., 2007*; *Rosa et al., 2011*). The classification of phagocytes into granulocytes and semigranulocytes could be helpful to further investigate the phagocytosis in the innate immune modulation and pathogen elimination of *C. gigas*.

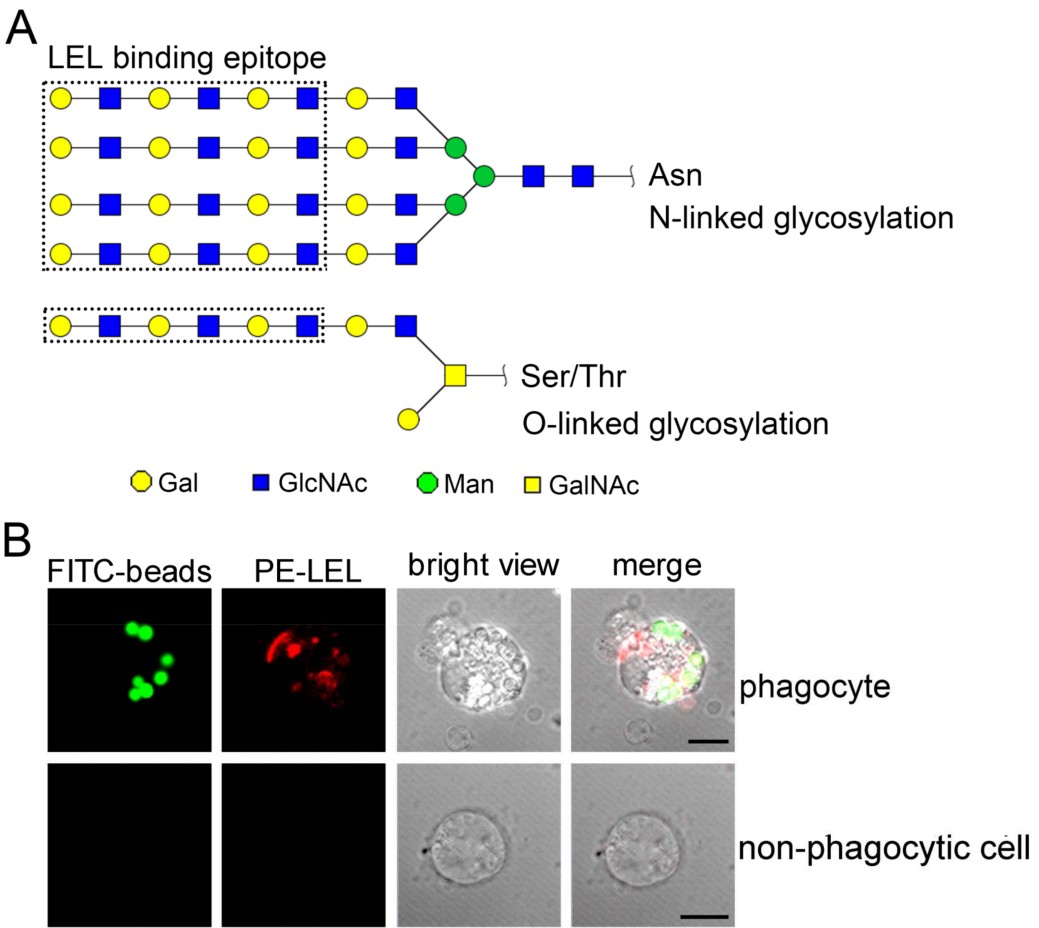

**Figure 7** **The distribution of polylactosamine in *C. gigas* haemocytes indicated by LEL staining.** LEL binding epitopes polylactosamine are indicated by dotted rectangles, and the predicted carbohydrate structures are represented in *N*-linked and *O*-linked glycans (A). The distribution of polylactosamine in oyster phagocytes revealed by confocal microscopy (B). Haemocytes were incubated with FITC-labeled latex beads, and then fixed and permeabilized, followed by PE-labeled LEL staining. A representative phagocyte and non-phagocyte are shown.

Molecular markers have been proved to be extremely useful for cell typing in mammalian immune system, while the cell typing of haemocytes in invertebrates still needs to be elucidated (*Kurucz et al., 2007*; *Le Foll et al., 2010*). Even if the invertebrate haemocytes are of great heterogeneity, a relationship has been found between different molecular markers and the corresponding cellular functions. In *Mytilus edulis*, the small granules of the granulocytes were found to be HPA-positive, and the large granules of the granulocytes were of WGA positive, indicating that lectin staining could be applied in the cell typing (*Pipe, 1990*). In *Crassostrea virginica*, haemocytes infected with *Haplosporidium nelsoni* (MSX) could be agglutinated by WGA and HPA, suggesting that haemocytes contained surface receptors resembling N-acetyl-D-glucosamine and $\alpha$-methylmannopyranoside (*Kanaley & Ford, 1990*). In *Anopheles gambiae*, only the granulocytes from individuals challenged by *Plasmodium falciparum* malaria could be stained by *Lens culinaris* agglutinin, whereas

most (96%) of naive granulocytes were negative and 4% were stained weakly ($p < 0.0001$) (*Rodrigues et al., 2010*). In the present study, the phagocytes could be separated from non-phagocytic cells based on the differential distribution of glycans by LEL staining. LEL exhibits high binding specificity towards polylactosamines with at least three lactosamine repeats, and has been widely used to discriminate different cell types (*Togayachi et al., 2007*). Most of the LEL positive haemocytes from *C. gigas* were phagocytes, suggesting the abundant distribution of polylactosamine glycans in phagocytes. Conversely the positive signals of WGA and PNA were observed in both phagocytes and non-phagocytic cells, indicating that there were no distribution differences of WGA- and PNA-binding epitopes between phagocytes and non-phagocytic cells in *C. gigas*.

Co-evolutionary arms races between pathogens and hosts, and the competitions are considered to be of immense importance in the evolution of living organisms (*Akira, Takeda & Kaisho, 2001*; *Akira, Uematsu & Takeuchi, 2006*). Integrins are required for cell-microorganism interaction and correct formation of phagosomes, and they play important roles in the phagocytosis (*Ballarin et al., 2002*; *Oliva et al., 2008*). In the present study, the blockage of $\beta$-integrin by RGD inhibited phagocytosis towards both Gram-negative and Gram-positive bacteria in granulocytes and semigranulocytes, indicating that $\beta$-integrin was extensively involved in the phagocytosis of oyster haemocytes. RGD-containing peptides were reported to induce haemocyte apoptosis in *C. gigas* at the concentration of 3 mM (*Terahara, Takahashi & Mori, 2003*; *Terahara, Takahashi & Mori, 2005*), which was much higher than that used in the present study, suggesting that lower concentration of RGD peptide could inhibit phagocytosis, while higher concentration of RGD peptide could induce cell apoptosis.

Lectins, typical pattern recognition receptors (PRRs), are involved in the pathogen recognition and phagocytosis. For example, a C-type lectin (CfLec-3) from *Chlamys farreri* with three carbohydrate-recognition domains (CRDs) could modulate haemocyte phagocytosis via binding to different PAMPs and microbes (*Yang et al., 2015*). The native lectin FcLec4 could bind to $\beta$-integrin to promote haemocytic phagocytosis in *Fenneropenaeus chinensis* (*Wang, Zhao & Wang, 2014*). Genes encoding lectin-like receptors (LLRs) are highly over-represented in oyster genome ($p < 0.0001$) (*Zhang et al., 2012*). In the present study, the blockage of LLRs by Fuc, Man and GlcNAc exhibited an inhibitory effect on phagocytosis of total haemocytes towards *V. splendidus* and *S. aureus*, indicating that LLRs were involved in the recognition of different microbes and modulation of phagocytosis. However, Glu exhibited little inhibitory effect on the phagocytosis towards *V. splendidus* or *S. aureus*, suggesting that the Glu binding LLRs might not participate in the phagocytosis of the two bacteria.

PAMPs are important stimuli in the activation of immune responses (*Iliev et al., 2005*). LPS has been proved to act as an extremely strong stimulator of innate immunity in mammals (*Alexander & Rietschel, 2001*; *Kawai & Akira, 2010*). The extracellular membrane receptors, such as Toll-like receptor 4 (TLR4), could recognize LPS and initiate the rapid immune-activation through an intracellular signaling pathway (*Chu & Mazmanian, 2013*; *Shenoy et al., 2012*; *West, Shadel & Ghosh, 2011*). Moreover, various forms of $\beta$-glucans have been proved to possess a potential value in shrimp and fish aquaculture, as they

could increase the numbers of circulating haemocytes, promote long-term activation of haemocytes and enhance the haemocytic aggregation (*Anderson et al., 2011*). In the present study, LPS stimulation substantially increased the phagocytic activity of both granulocytes and semigranulocytes towards *V. splendidus* and *S. aureus*, while PGN stimulation had no effect on the phagocytosis. It is noteworthy that a number of Gram-negative bacteria, including *V. splendidus*, have been identified to be important aquaculture pathogens, which could cause massive mortalities of oysters (*Garnier, Labreuche & Nicolas, 2008*; *Richards et al., 2015*). The enhancement of phagocytic activity towards microbial pathogens under LPS stimulation contributed to better understand the modulation of phagocytosis in oysters, and suggested the potential application in oyster aquaculture.

In conclusion, the present study showed that *C. gigas* haemocytes can be categorised into three cell types including granulocytes, semigranulocytes and agranulocytes. The phagocytic capacity of granulocytes and semigranulocytes towards different microbes was determined, and the $\beta$-integrin and certain LLRs were found to play important roles in the phagocytosis of granulocytes and semigranulocytes. In addition, LPS but not PGN could significantly enhance the phagocytic activities. Moreover, LEL binding epitope polylactosamine was highly distributed on the extracellular cell surface of phagocytes, which could represent a potential molecular marker to differentiate phagocytes from non-phagocytic haemocytes.

## ACKNOWLEDGEMENTS

The authors would like to thank Dahai Gao (Institute of Oceanology, Chinese Academy of Sciences) for the support in the flow cytometry and Sheng Wang (Sun Yat-sen University) for the help in the cell staining.

### Funding

This research was supported by the Natural Science Foundation of China (Nos. 31530069, 41406170), the Research Foundation for Talented Scholars in Dalian Ocean University (to LS), and an earmarked fund (CARS-48) for Modern Agro-industry Technology Research System. The funders had no role in study design, data collection and analysis, decision to publish, or preparation of the manuscript.

### Grant Disclosures

The following grant information was disclosed by the authors:
Natural Science Foundation of China: 31530069, 41406170.
Research Foundation for Talented Scholars in Dalian Ocean University.
Modern Agro-industry Technology Research System: CARS-48.

### Competing Interests

Linsheng Song and Lingling Wang are Academic Editors for PeerJ.

## Author Contributions

- Shuai Jiang conceived and designed the experiments, performed the experiments, analyzed the data, wrote the paper, prepared figures and/or tables, reviewed drafts of the paper.
- Zhihao Jia and Tao Zhang performed the experiments, contributed reagents/materials/analysis tools.
- Lingling Wang and Linsheng Song conceived and designed the experiments, wrote the paper, prepared figures and/or tables, reviewed drafts of the paper.
- Limei Qiu analyzed the data, contributed reagents/materials/analysis tools.
- Jinsheng Sun analyzed the data.

## Data Availability

The raw data has been supplied as a Supplementary File.

## Supplemental Information

Supplemental information for this article can be found online at http://dx.doi.org/10.7717/peerj.2590#supplemental-information.

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
