# Peer review of "Functional characterisation of phagocytes in the Pacific oyster Crassostrea gigas"

_PeerJ, doi:10.7717/peerj.2590_

## Round 0.1 · original submission · Major Revisions

The paper has to be modified according to the suggestions given by the three reviewers. Furthermore the English should be corrected by a native speaker.

·

Basic reporting

-Clear, unambiguous, professional English language used throughout.
The English language is not always clear and unambiguous; some periods (abstract) that need language revision are highlighted in yellow.

-Figures are relevant, high quality, well labelled & described.
It is possible to improve the quality of granular and semigranular cells in fig 1A? The different population are evident in the flow cytometric analysis, not so evident the difference from cell stain.

Experimental design

No Comments

Validity of the findings

No Comments

Additional comments

Some sentences (mainly in the abstract) need to be claryfied. All my questions are reported in the PDF of the manuscript as comments

Reviewer 2 ·

Basic reporting

The research appears very interesting because for the first time the Oyster hemocytes have been clearly separed and biologically characterized into three subpopulations by flow cytometry. The authors also given information about the different ability to engulf FITC-labelled latex beads and different microbes. Additional phagocytosis analysis including lectin and LPS interaction could improve the knowledge about pathogen aquaculture studies. The results represent an advance in the understanding the immune functions of hemocyte phagocytosis but at the present state of the paper is acceptable for publication after major revision.
The article should not include sufficient recent information, i.e. a long paragraph introducing mammals phagocytosis following a not evolutionary and invertebrates history of the process. Relevant and more recent literature should be appropriately included.
Figures are relevant to the content of the article and appropriately described and, in general is near to be appropriate ‘unit of publication’.

Experimental design

The submission is an original primary research
The investigation has been conducted rigorously and to a high technical standard and innovative standard as flow cytometry and confocal microscopy.

Validity of the findings

The data should be robust, and controlled but some modification, including the statistical analysis, reported below are necessary:
INTRODUCTION
Rows 51-57 synthetize,
Row 59 The word “higher animal” is empty of content
Row 72 – 81 unclear and in a incorrect position
MATERIALS AND METHODS
Rows 97-106, please include the number of cell per animal and obtained at the end of preparation.
Rows 105 -106 The author MUST be demonstrated that the cell pool not modified the cell state by additional experiment or specific literature references.
RESULTS
Due to the high variability showed in the figure between homolog experiments, the data should be analyzed by using ANOVA and/or similar non parametric tests.
Row 189 do not fit with the number of hemocytes cited I suggest the use of the “fold” respect to the control. In this case the use of percentage especially for activation data (>100%) could be a problem for the selection of statistic tests
the related figure

Reviewer 3 ·

Basic reporting

English spelling should be checked. Figures are relevant, with the exception of figure 3, the left part of which should be deleted to render it simpler. One of the reference is a duplicate of the previous one. Methods are appropriate and well described. Some of the results in the figures are not reported in the result section and in discussion (e.g,., merge in fig. 7B)

Experimental design

The manuscript reports a characterization of the haemocytes of the phagocytes of the Pacific oyster Crassostrea gigas. Using both light microscopy and FACS, they divided haemocytes in agranulocytes, semigranulocytes and granulocytes, and could state that semigranulocytes and granulocytes are the cells with phagocytic activity. They also studied the effects of sugars on phagocytosis and the role of integrins and demonstrated that the lectin from Lycopersicon esculentum recognized only phagocytes no other cells.
Methods are appropriate and well described. The experimental design, although good, suffers from some inconsistencies, such as the use of different target particles in light microscopy and FACS analysis.

Validity of the findings

Collectively, the data are robust and statistically controlled. The results will be useful to researchers involved in the study of stress responses in oysters.
The manuscript suffers from some incorrect expression in the text. Some major and minor comments are reported in the enclosed text.

Annotated reviews are not available for download in order to protect the identity of reviewers who chose to remain anonymous.

---

## Round 0.2 · Minor Revisions

The manuscript has greatly improved following the referees' suggestions. However, some additional changes are required, as indicated in the enclosed file from Reviewer 3.

·

Basic reporting

No Comments

Experimental design

No Comments

Validity of the findings

No Comments

Additional comments

I carefully read the revised manuscript entitled “Functional characterisation of phagocytes in the Pacific oyster Crassostrea gigas” and I have verified that all questions have been satisfactorily answered. Therefore, in my opinion, it is acceptable for publication in its present form.

Reviewer 2 ·

Basic reporting

Basic reporting

The research appears very interesting because for the first time the Oyster hemocytes have been clearly separed and biologically characterized into three subpopulations by flow cytometry. The authors also given information about the different ability to engulf FITC-labelled latex beads and different microbes. Additional phagocytosis analysis including lectin and LPS interaction could improve the knowledge about pathogen aquaculture studies. The results represent an advance in the understanding the immune functions of hemocyte phagocytosis
Figures are relevant to the content of the article and appropriately described and, in general is near to be appropriate ‘unit of publication’.

Experimental design

The submission is an original primary research
The investigation has been conducted rigorously and to a high technical standard and innovative standard as flow cytometry and confocal microscopy.

Validity of the findings

The data should be robust, and controlled

Additional comments

All th suggestions and answers in REV1 have been solved by the authors

Reviewer 3 ·

Basic reporting

The manuscript has greatly improved following the referees' suggestions. However, some additional changes are required, as indicated in the enclosed file.
In addition, although the manuscript has been revised once, the English spelling is not yet fully correct. In particular, authors, as already suggested, should use British English instead of a mixture of American and British English.

Experimental design

The manuscript, in its present form, satisfies the requirements of the journal.

Validity of the findings

Findings are new and worthy to be published.

Additional comments

The manuscript still required additional revision.

Annotated reviews are not available for download in order to protect the identity of reviewers who chose to remain anonymous.

---

## Round 0.3 · accepted · Accept

The manuscript has been improved.